# Genotype-Phenotype Correlations in PMM2-CDG

**DOI:** 10.3390/genes12111658

**Published:** 2021-10-21

**Authors:** Laurien Vaes, Daisy Rymen, David Cassiman, Anna Ligezka, Nele Vanhoutvin, Dulce Quelhas, Eva Morava, Peter Witters

**Affiliations:** 1Faculty of Medicine, KU Leuven, B3000 Leuven, Belgium; laurien.vaes@student.kuleuven.be; 2Center for Metabolic Diseases, Department of Paediatrics, University Hospitals Leuven, B3000 Leuven, Belgium; daisy.rymen@uzleuven.be (D.R.); nele.vanhoutvin@uzleuven.be (N.V.); 3Department of Gastroenterology-Hepatology, University Hospitals Leuven, B3000 Leuven, Belgium; David.cassiman@uzleuven.be; 4Department of Clinical Genomics, and Department of Laboratory Medicine and Pathology, Mayo Clinic, Rochester, MN 55902, USA; Ligezka.anna@mayo.edu (A.L.); Morava-kozicz.eva@mayo.edu (E.M.); 5Unidade de Bioquímica Genética, Centro de Genética Médica, Centro Hospitalar Universitário do Porto, 4099-001 Porto, Portugal; mdquelhas@gmail.com; 6Unit for Multidisciplinary Research in Biomedicine, Abel Salazar Biomedical Sciences Institute, University of Porto, 4050-313 Porto, Portugal; 7Centro Referência Doenças Hereditárias do Metabolismo, Centro Hospitalar Universitário do Porto, 4099-001 Porto, Portugal

**Keywords:** PMM2-CDG, NPCRS, congenital disorders of glycosylation, genotype, mutation

## Abstract

PMM2-CDG is a rare disease, causing hypoglycosylation of multiple proteins, hence preventing full functionality. So far, no direct genotype–phenotype correlations have been identified. We carried out a retrospective cohort study on 26 PMM2-CDG patients. We collected the identified genotype, as well as continuous variables indicating the disease severity (based on Nijmegen Pediatric CDG Rating Score or NPCRS) and dichotomous variables reflecting the patients’ phenotype. The phenotypic effects of patients’ genotype were studied using non-parametric and Chi-Square tests. Seventeen different pathogenic variants have been studied. Variants with zero enzyme activity had no significant impact on the Nijmegen score. Pathogenic variants involving the stabilization/folding domain have a significantly lower total NPCRS (*p* = 0.017): presence of the p.Cys241Ser mutation had a significantly lower subscore 1,3 and NPCRS (*p* = 0.04) and thus result in a less severe phenotype. On the other hand, variants involving the dimerization domain, p.Pro113Leu and p.Phe119Leu, resulted in a significantly higher NPCRS score (*p* = 0.002), which indicates a worse clinical course. These concepts give a better insight in the phenotypic prognosis of PMM2-CDG, according to their molecular base.

## 1. Introduction

PMM2-CDG, the most frequent type of congenital disorders of glycosylation, is caused by pathogenic variants in the *PMM2* gene leading to impaired activity of phosphomannomutase 2 (PMM2) [1], which catalyzes the conversion of mannose-6-phosphate into mannose-1-phosphate [2,3,4,5]. Deficiency of PMM2 eventually leads to hypoglycosylation of proteins, preventing them to obtain proper intra- and extracellular localization and full functionality [6]. The most common phenotypic features include developmental delay, ataxia, hypotonia, strabismus, feeding difficulties, inverted nipples and abnormal fat distribution [7,8]. In 96% of the PMM2-CDG population, developmental disability is observed [9].

*PMM2*, located on chromosome 16p13, is a small gene consisting of eight exons and encodes a protein of 246 amino acids [1,4]. Each chain of the homodimer is composed of 2 domains connected through hinge peptides, a cap domain (residues 1–81 and 189–247) and a core domain (residues 86–185), that require mannose or glucose for activation. After ligand binding, a conformational change creates an exclusive environment for catalysis [10,11]. Until now, 118 pathogenic mutations have been described in this gene, 93 are missense mutations [12]. Complete absence of PMM2 activity is found to be not compatible with life, hence most patients are compound heterozygous with one inactivating and one hypomorphic mutation [13].

Based on the identified X-ray crystal structure of PMM1, which is the PMM2 isozyme and has a 65% sequence identity, the impact of various mutations can be predicted [9,10]. They can be classified into three main groups according to their targeting site: substrate binding and catalysis; dimerization or protein stability [10]. The most common mutation, p.Arg141His, has an estimated carrier frequency of 1/79 in the Dutch population and an estimated frequency of 45–60% in the PMM2-CDG population [9,14]. It leads to binding of the bisphosphate sugar followed by domain closure resulting in inactivation of the protein [9,13,14,15].

Unfortunately, no clear correlations between the genotype and phenotype in PMM2-CDG have been described yet as the phenotypic spectrum is extremely variable affecting multiple systems [16]. The clinical presentation can range from asymptomatic to potential lethal conditions as well as an intrafamilial variability between siblings with the same genotype is seen [17,18,19]. Some studies indicate there might be a link between the disease severity and the genotype and that mutations in *PMM2* affecting the folding or stability seem to be associated with a milder phenotype [9]. First of all, Matthijs et al. found that the [p.Asp188Gly] + [p.Arg141His] genotype resulted in a severe phenotype with a high mortality [20,21]. Secondly, a mild phenotype seems to be associated with pathogenic genotypic variants located on the C-terminal part of the protein such as p.His218Leu, p.Thr237Met, or p.Cys241Ser [1]. This mild phenotypic association with the p.Cys241Ser mutation was confirmed in other studies as well [22,23]. A more severe association was observed with the most common genotype in PMM2-CDG, [p.Arg141His] + [p.Phe119Leu] [24].

In this retrospective study, the genotype and phenotype features of 26 proven PMM2-CDG patients have been evaluated to identify correlations. The aim of this study is to get a better insight in the prognosis and possible outcome.

## 2. Materials and Methods

### 2.1. Study Design and Population

We conducted a retrospective cohort study of 26 patients, who are genetically confirmed PMM2-CDG patients carrying two identified known-pathogenic variants in *PMM2*. The patients were all followed/observed in two academic centers (University Hospitals Leuven, Leuven, Belgium and Mayo Clinic, Rochester, MN, USA) between 1 April 2018 and 30 August 2020. The retrospective data collection of the patients from Mayo clinic is part of the natural history study (IRB19-005187, Clinical and Basic Investigations into Congenital Disorders of Glycosylation). Ethical approval was obtained from the Research Ethics Committees UZ/KU Leuven (OBC) with MP014902 as reference number. All PMM2-CDG patients being monitored in these centers were eligible for study inclusion: no patients were excluded. This resulted in inclusion of twelve patients originating from Mayo Clinic, Rochester, USA and fourteen patients followed in UZ Leuven located in Leuven, Belgium. The data must be considered as pseudo-randomized, because only people with PMM2-CDG were eligible.

### 2.2. Variables

All variables were extracted from the patients’ medical files and summarized to get a detailed overview. To protect the participants’ privacy, the data was anonymized. The genotype of all patients, serving as independent variable, was already identified through molecular testing. The Nijmegen pediatric CDG rating scale (NPCRS) served as primary dependent variable of interest. This rating scale was developed in 2011 by Achouitar et al. and comprises a questionnaire filled out by clinical experts to estimate the clinical severity of CDG [25]. The Nijmegen CDG severity score is a standardized questionnaire with a proven good inter-rater comparison, as well as a stable score gradient of the clinical course in long-term follow-up studies [8]. The total Nijmegen score is obtained by summation of the subscores from three different sections. Section 1 evaluates five general functions (vision, hearing, communication, feeding and mobility). In Section 2, the system-specific involvement over the preceding 6 months is quantified (CNS, blood, gastro-intestinal, endocrine, respiratory, cardiovascular, renal and liver function). Finally, Section 3 evaluates the Current Clinical Assessment and includes questions about growth, development, vision, strabismus, myopathy, ataxia, pyramidal symptoms, extrapyramidal symptoms and neuropathy. For each item in the questionnaire, there are four responses possible with increasing severity: normal (0), mild (1), moderate (2) and severe (3) [25]. So, a high total score indicates a severe clinical course of the disease.

Furthermore, basic patient characteristics, such as gender or age at evaluation, along with other rarer phenotypic features, such as pericardial effusion, are included as dependent variables. The more specific phenotypic features are classified according to their system involvement (neurological, congenital, ophthalmological, etc.) and transformed into binary data. Thereby they represent the presence or absence of a certain feature that is necessary to perform adequate statistical analysis further on.

### 2.3. Statistical Analysis

All analyses are performed using JMP software (JMP version 15.2.1 from SAS system for Windows). Mean and standard deviation are used to present descriptive statistics for continuous variables. Non-parametric tests, the Mann–Whitney U or Kruskal–Wallis test, are used for continuous variables to compare data between different groups. These tests are one-sided. Dichotomous variables, such as gender, are presented by the frequency of cases and with their respective percentages. Significant differences between groups, regarding dichotomous variables, are analyzed using the Chi-square test, which represents a one-sided test. False discovery rate (FDR) is used when conducting multiple comparisons to prevent false positive outcomes. For all tests, a significance level of 5% is assumed.

## 3. Results

### 3.1. Patient Characteristics

Twenty-five patients were compound heterozygous, one was homozygous for p.Phe119Leu. The distribution of the genotype and their perspective percentages is given in Table 1, while Figure 1 illustrates the patients’ pathogenic variants according to their targeting site. The targeting site of the nucleotide changes is used to sort different groups for analysis. The most frequent mutation, p.Arg141His, was found in 18 patients (69%), of whom 7 patients (26.9%) had the [p.Arg141His] + [p.Phe119Leu] genotype. The youngest patient included was 8 months old, while the oldest patient was 45 years. A mean age of 13 years 9 months old and a standard deviation of 13 years and 3 months was calculated. 58% of the population were male, 42% were female. No significant difference in Nijmegen subscores or total NPCRS score (*p*-value 0.86) was noted between males and females. All patients underwent a sialotransferrin analysis, all but one, bearing the known pathogenic variants [p.Cys241Ser] + [p.Arg141His], had an abnormal sialotransferrin pattern (type I pattern).

### 3.2. Pathogenic Variants with No Residual Enzyme Activity

Three pathogenic variants, p.Arg141His, p.Thr237Arg and p.Asp188Gly, are known to express no or very limited residual enzyme activity (0–1.88%) [15,18,24]. We analyzed if these changes, resulting in a non-functioning allele of *PMM2*, carry a significant effect on the clinical course. All patients were divided into two groups, representing patients containing one non-functional variant (*n* = 22) on the one hand and patients composite with two functioning variants (*n* = 4) on the other hand. Table 2 displays the average and standard deviation of the different Nijmegen subscores and total NPCRS score and their respective *p*-value. No significant difference in gender distribution (*p*-value 0.57) between the two groups was found. Comparison of the total subscores and total NPCRS score (*p*-value 0.79) showed no significant results. Only 21 out of 26 patients were ultimately included in the non-parametric test of the total score of NPCRS due to missing data of 5 patients. Next, we wanted to evaluate the role of the second pathogenic variant in the presence of one non-functioning allele, hence the four patients, belonging to group 2, meaning they have two functioning alleles, will be excluded in the analyses performed. This exclusion will lead to the comparison of congruent groups who all have one non-functioning allele, whereby the phenotypical effect of the second variant present will be magnified and better evaluated. A false discovery rate (FDR) corrected Chi-square test yielded no significant correlations between specific phenotypic features and the presence of one non-functioning allele with a presumed 5% significance level.

### 3.3. Nucleotide Changes Involving the Folding or Stabilization Domain

Six pathogenic variants are acknowledged to affect the folding or stabilization domain of the PMM2 homodimer (see Figure 1): p.Val231Met, p.Cys241Ser, p.Asp148Asn, p.Arg162Trp, p.Phe183Ser and p.Ile132Thr [10,15,20,26]. Ten patients contained one folding and/or a stabilization variant, eight out of ten patients were composite with one non-functioning allele (p.Arg141His, p. Thr237Arg or p.Asp188Gly). Two patients were composite [p.Asp148Asn] + [p.Pro113Leu] and [p.Arg162Trp] + [c.451_454del]. To perform adequate analysis, these last two patients were excluded and only patients with one non-functioning allele were included. This will give us a better insight in the phenotypic outcome of nucleotide changes involving folding or stabilization as the second variant remains the same between the two studied groups. Results of the statistical comparison between patients with a known folding or stabilization nucleotide change combined with one non-functioning allele (*n* = 8) and patients with a non-functioning allele combined with another allele (*n* = 14) are shown in Table 3. Patients with a nucleotide change involving folding or stabilization have a significant lower total subscore 1 (*p*-value 0.02), 3 (*p*-value 0.01) and total NPCRS score (*p*-value 0.04). No significant correlations between specific phenotypic features and the presence of an allele with one folding or stabilization variant was found using FDR-corrected Chi-square test.

Three patients had a common pathogenic variant, p.Cys241Ser, all three combined with one non-functioning allele (p.Arg141His or p.Thr237Arg). We compared the different scores between patients who had an inactivating mutation combined with p.Cys241Ser (*n* = 3) and patients with one inactivating mutation (*n* = 19). Non-parametric testing, as illustrated in Table 4, showed a significant difference between the two groups regarding total subscore 1 (*p*-value 0.02), 2 (*p*-value 0.03) and total NPCRS score (*p*-value 0.04). Patients with the p.Cys241Ser variant have a significant lower total subscore 1, 2 and total NPCRS compared to other heterozygous patients with one non-functioning allele. Significant results are also seen when comparing p.Cys241Ser combined with a non-functioning allele (*n* = 3), with other variants involved in folding or stabilization domain combined with a non-functioning allele (*n* = 5). The presence of p.Cys241Ser results in a significant lower total subscore 1 (*p*-value 0.03) and total subscore 3 (*p*-value 0.02). No significant lower total NPCRS is seen between the two compared groups (*p*-value 0.06).

### 3.4. Nucleotide Changes Involving the Dimerization Domain

Finally, patients were divided into two groups based on whether the pathogenic variant affects the dimerization domain. In our study population, 13 patients contained mutations affecting the dimerization domain p.Pro113Leu or p.Phe119Leu. 11 out of these 13 patients were compound heterozygous with one inactive pathogenic variant. One was homozygous [p.Phe119Leu] + [p.Phe119Leu] and the last one was compound heterozygous [p.Asp148Asn] + [p.Pro113Leu]. To compare equal groups, the first group (*n* = 11) was composed of heterozygous patients containing one inactive pathogenic variant (p.Arg141His or p.Thr237Arg) and one pathogenic variant involving the dimerization domain. The second group comprised patients with one non-functioning allele (p.Arg141His or p.Thr237Arg). The results are shown in Table 5. A significant difference was seen in total subscore 1 (*p*-value 0.001), 3 (*p*-value 0.004) and the total NPCRS score (*p*-value 0.002). Patients with a pathogenic variant in the dimerization domain have a significant higher total subscore 1, 3 and total NPCRS score. A FDR-corrected Chi-square test revealed a significant difference in the presence of a gastric feeding tube between the two groups (*p*-value 0.005), whereby patients with a dimerization variant seem to have a higher chance in needing a gastric feeding tube.

## 4. Discussion

The aim of this study was to reveal unknown genotype-phenotype correlations to get a better insight in the clinical prognosis of patients. As described in the introduction, literature review did not reveal proven direct associations between genotype and phenotype in PMM2-CDG, hence this study is the first that aims to describe them. Three important findings are revealed/exposed in this study. First of all, no significant difference is seen between patients composite with or without one inactive *PMM2* variant (p.Arg141His, p.Thr237Arg or p.Asp188Gly). This indicates that the PMM2 phenotype is most likely determined by the second pathogenic variant present in PMM2-CDG patients and that the second allele must be a milder mutation, as most patients were compound heterozygous with one loss of function mutation and one hypomorphic mutation as a complete absence of PMM2 activity is found to be incompatible with life [13,21]. The latter is confirmed also in our data as no homozygous state for p.Arg141His variant has been identified in patients, while according to the Hardy-Weinberg equilibrium, homozygous p.Arg141His genotype should be expected in 45% to 60% of the PMM2-CDG population as this mutation is the most common mutation found [14]. Loss of function mutations are most generally found to be recessive, whereby the single functioning allele is able to provide enough gene product to produce a viable phenotype [27]. Secondly, Altassan et al. already speculated that mutations affecting the folding and/or stability domain of PMM2 were associated with a milder phenotype [9]. Non-parametric tests performed in this study confirm this association. Patients compound heterozygous for one non-functioning allele combined with a folding and or stabilization mutation, p.Val231Met, p.Cys241Ser, p.Asp148Asn, p.Arg162Trp, p.Phe183Ser and p.Ile132Thr, have a significant lower subscore 1, 3 and total NPCRS. Detailed analysis of system-specific involvement, which showed no significant difference between groups, does not result in any significant correlations. p.Cys241Ser has a significant lower score, which suggests this is one of the best prognostic mutations in PMM2-CDG. Lastly, the fact that compound heterozygous patients with one inactivating variant and one pathogenic variant affecting the dimerization domain, p.Pro113Leu and p.Phe119Leu, have a significant higher total subscore 1 (five general functions), 3 (Current Clinical Assessment) and total NPCRS score reflects a more severe clinical course of PMM2-CDG. Subscore 2 (system-specific involvement) showed no significant higher score. Therefore, this analysis mostly captures the neurologic phenotype. System-specific involvement was then further analyzed using FDR-corrected Chi-Square test. Only one direct correlation was observed between having a gastric feeding tube and presence of nucleotide changes in the dimerization domain. We can presume that other nucleotide changes affecting the dimerization domain, such as p.Thr118Ser, combined with an inactive variant would result in a similarly severe clinical course. According to the in-silico analysis, this latter nucleotide change p.Thr118Ser, was predicted to be a milder mutation with a 54% residual enzyme activity [15].

Fifteen nucleotide changes observed in this study were reported by the Human Genome Mutation Database. Two variants, p.Gln33Pro and p.Ser47Leu, were only found recently [12,28]. This study does not contain any variants of unknown significance. All pathogenic variants were missense mutations, except for one being a short deletion in exon 6 resulting in a frameshift mutation. Variants can be classified according to their effect on different functional domains (see Figure 1): two variants compromising the catalytic site (p.Arg141His, p.Asp188Gly), two pathogenic variants disrupting the dimer interface (p.Pro113Leu and p.Phe119Leu) and four misfolding changes (p.Arg162Trp, p. Val231Met, p.Thr237Arg and p.Cys241Ser) were present in our cohort [10,15,20]. p.Asp148Asn is known to produce a thermolabile protein, while p.Phe183Ser and p.Ile132Thr are predicted to decrease stability [15,26]. P.Glu139Lys is found to disrupt the splicing enhancer sequence resulting in exon skipping [29]. The effect of numerous nucleotide changes is yet to be determined (p.Ile153Thr, c.451_454del, p.Gln33Pro, p.Ser47Leu, p.Thr18Ser). The p.Arg141His allele had the highest prevalence in the study cohort (69%), which is comparable with frequencies seen in other study populations with a p.Arg141His allele frequency ranging from 40–61% [21,23]. The most frequent genotype is known to be [p.Arg141His] + [p.Phe119Leu], this genotype has a frequency of 26,9% in our cohort [21].

These new findings are a helpful tool for clinicians to give patients a more adequate prediction of their clinical prognosis, as well as it emphasizes a closer follow-up in patients with a variant in the dimerization domain as it seems to result in a more severe clinical outcome.

From three patients, no total subscore 1 was obtained due to missing data. These patients were aged less than one year, whereby a score for self-education and educational achievement could not be assigned to these patients and thereby resulting in no total subscore 1 and total NPCRS score. These patients were composite [p.Arg141His] + [p.Val231Met]; [p.Phe119Leu] + [p.Arg141His] and [p.Phe119Leu] + [p.Arg141His]. Two patients had one missing data in subscore 3, therefore no total subscore 3 and total NPCRS score could be calculated for these patients. These two adult patients had a lack of score for the category development and were composite [p.Arg141His] + [p.Cys241Ser] and [p.Arg162Trp] + c.451_454del.

PMM2-CDG is a rare disease, and therefore it can be presumed that a sample size of 26 patients is sufficient to perform the study [21]. Looking at the different non-parametric tests performed, the small sample size in the subgroups remains a limitation of this study, especially when evaluating the p.Cys241Ser variant (*n* = 3). We recommend that this study should be rerun when more patients can be included. We also propose to include other parameters, such as genes or environmental influences to study intrafamilial variability.

## 5. Conclusions

Until now, no direct correlations between the genotype and phenotype of PMM2-CDG have yet been described. We conclude that mutations affecting the folding or stabilization domain have a significantly lower total NPCRS score and thus provide a good prognostic clinical outcome. While nucleotide changes in the dimerization interface, p.Pro113Leu and p.Phe119Leu, have a significantly higher total NPCRS which indicates a more severe clinical outcome. These findings can have implications regarding new pathogenic variants found in the dimerization or stabilization/folding domain as we suspect a similar severity in clinical course.

## Figures and Tables

**Figure 1 genes-12-01658-f001:**
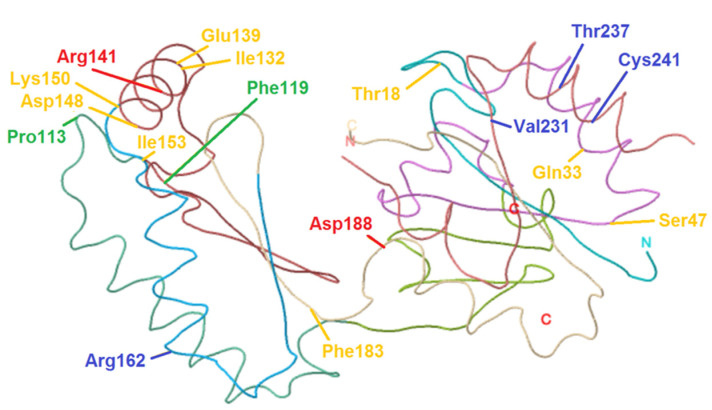
Structure of PMM2 protein and the changes described in this study. Residues involved in the catalytic site are indicated in red, dimerization residues are highlighted in green and lastly, folding residues are depicted in blue. The remaining residues, highlighted in yellow, could not be classified into these previous groups as their function is not fully known or they have a different functioning mechanism.

**Table 1 genes-12-01658-t001:** Distribution of the different pathogenic variants found in all 26 patients and their perspective percentages in parentheses.

First Pathogenic Variant	Second Pathogenic Variant	Number of Patients (*N* = 26)
p.Arg141His		
	p.Phe119Leu	7 (26.9%)
	p.Cys241Ser	2 (7.7%)
	p.Pro113Leu	2 (7.7%)
	p.Val231Met	2 (7.7%)
	p.Glu139Lys	1 (3.8%)
	p.Thr18Ser	1 (3.8%)
	p.Phe183Ser	1 (3.8%)
	p.Ile132Thr	1 (3.8%)
	p.Ile153Thr	1 (3.8%)
p.Thr237Arg		
	p.Pro113Leu	2 (7.7%)
	p.Cys241Ser	1 (3.8%)
p.Gln33Pro		
	p.Ser47Leu	1 (3.8%)
p.Phe119Leu		
	p.Phe119Leu	1 (3.8%)
p.Arg162Trp		
	c.451_454del (p.Glu151Ilefs*2)	1 (3.8%)
p.Asp188Gly		
	p.Val231Met	1 (3.8%)
p.Asp148Asn		
	p.Pro113Leu	1 (3.8%)

**Table 2 genes-12-01658-t002:** Subscores and total Nijmegen score in patients with one non-functioning allele and patients with other nucleotide changes.

Variables	*N* ^1^	Group 1 ^2^ *N* = 22	Group 2 ^3^ *N* = 4	*p*-Value
**Gender**	26			0.5739
**Female**		9 (34.6%)	2 (7.69%)	
**Male**		13 (50.00%)	2 (7.69%)	
**Total subscore 1**	23	9.45 (+/− 5.20)	8.25 (+/− 1.26)	0.4623
**Total subscore 2**	26	3.77 (+/− 2.52)	3.50 (+/− 1.73)	0.7189
**Total subscore 3**	24	11.70 (+/− 5.01)	12.50 (+/− 1.91)	0.4601
**Total score NPCRS**	21	25.97 (+/− 11.35)	24.25 (+/− 3.10)	0.7880

Abbreviation: NPCRS, Nijmegen Pediatric CDG Rating Scale. Dichotomous variable (gender) is expressed as number of frequencies with percentage in parentheses. Continuous variables are expressed in their mean score with their perspective standard deviation in parentheses. *p*-values, reported in this table, are two-sided. ^1^ Illustrates the total number of responses per variable. ^2^ Represents patients containing one allele leading to a null residual enzyme activity (p.Arg141His, p.Thr237Arg or p.Asp188Gly). ^3^ Represents all patients with two functioning pathogenic variants.

**Table 3 genes-12-01658-t003:** Subscores and total Nijmegen score in patients with nucleotide change involving folding or stabilization together with one non-functioning allele and patients with one non-functioning allele combined with another allele.

Variables	*N* ^1^	Group 1 ^2^ *N* = 8	Group 2 ^3^ *N* = 14	*p*-Value
**Gender**	22			0.5836
**Female**		3 (13.64%)	6 (27.27%)	
**Male**		5 (22.73%)	8 (36.36%)	
**Total subscore 1**	19	6.29 (+/− 4.42)	11.29 (+/− 4.85)	**0.0196**
**Total subscore 2**	22	3.50 (+/− 2.73)	3.93 (+/− 2.50)	0.7292
**Total subscore 3**	20	8.14 (+/− 3.24)	13.62 (+/− 4.81)	**0.0123**
**Total score NPCRS**	17	19.33 (+/− 9.65)	29.59 (+/− 10.91)	**0.0444**

Abbreviation: NPCRS, Nijmegen Pediatric CDG Rating Scale. Dichotomous variable (gender) is expressed as number of frequencies with percentage in parentheses. Continuous variables are expressed in their mean score with their perspective standard deviation in parentheses. All *p*-values, reported in this table, are two-sided. ^1^ Indicates the total number of responses per variable. ^2^ Represents patients composite with one folding or stabilization nucleotide change combined with one inactivating mutation (p.Arg141His, p.Thr237Arg). ^3^ Represents all patients with one inactivating mutation (p.Arg141His, p.Thr237Arg or p.Asp188Gly). Bold value: *p* < 0.05.

**Table 4 genes-12-01658-t004:** Subscores and total Nijmegen score in patients with p.Cys241Ser together with one non-functioning allele and patients with one non-functioning allele.

Variables	*N* ^1^	Group 1 ^2^ *N* = 3	Group 2 ^3^ *N* = 19	*p*-Value
**Gender**	22			0.6416
**Female**		1 (4.55%)	8 (36.36%)	
**Male**		2 (9.09%)	11 (50.00%)	
**Total subscore 1**	19	2.67 (+/− 2.50)	10.72 (+/− 4.56)	**0.0158**
**Total subscore 2**	22	0.67 (+/− 0.58)	4.26 (+/− 2.35)	**0.0290**
**Total subscore 3**	20	5.50 (+/− 4.95)	12.39 (+/− 4.64)	0.0879
**Total score NPCRS**	17	10.00 (+/− 7.07)	28.10 (+/− 10.12)	**0.0369**

Abbreviation: NPCRS, Nijmegen Pediatric CDG Rating Scale. Dichotomous variable (gender) is expressed as number of frequencies with percentage in parentheses. Continuous variables are expressed in their mean score with their perspective standard deviation in parentheses. All *p*-values, reported in this table, are two-sided. ^1^ Indicates the total number of responses per variable. ^2^ Represents patients composite p.Cys241Ser combined with one inactivating mutation (p.Arg141His, p.Thr237Arg). ^3^ Represents all patients with one inactivating mutation (p.Arg141His, p.Thr237Arg or p.Asp188Gly). Bold value: *p* < 0.05.

**Table 5 genes-12-01658-t005:** Subscores and total Nijmegen score in patients with one variant affecting dimerization domain together with one non-functioning allele and patients with one non-functioning allele.

Variables	*N* ^1^	Group 1 ^2^ *N* = 11	Group 2 ^3^ *N* = 11	*p*-Value
**Gender**	22			0.5000
**Female**		4 (18.18%)	5 (22.73%)	
**Male**		7 (31.82%)	6 (27.27%)	
**Total subscore 1**	19	13.17 (+/− 3.22)	6.10 (+/− 4.31)	**0.0012**
**Total subscore 2**	22	4.55 (+/− 2.38)	3.00 (+/− 2.35)	0.1428
**Total subscore 3**	20	14.70 (+/− 4.81)	8.70 (+/− 3.13)	**0.0039**
**Total score NPCRS**	17	34.19 (+/− 7.60)	18.67 (+/− 8.93)	**0.0021**

Abbreviation: NPCRS, Nijmegen Pediatric CDG Rating Scale. Dichotomous variable (gender) is expressed as number of frequencies with percentage in parentheses. Continuous variables are expressed in their mean score with their perspective standard deviation in parentheses. All *p*-values, reported in this table, are two-sided. ^1^ Indicates the total number of responses per variable. ^2^ Represents patients composite heterozygous dimerization variant combined with one inactivating mutation (p.Arg141His or p.Thr237Arg). ^3^ Represents all patients with one inactivating mutation (p.Arg141His, p.Thr237Arg or p.Asp188Gly). Bold value: *p* < 0.05.

## Data Availability

The data presented in this study are available on request from the corresponding author.

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
