# Peer review of "Genotype-Phenotype Correlations in PMM2-CDG"

_genes, 2021, doi:10.3390/genes12111658_

Round 1

Reviewer 1 Report

The article from Vaes et al. reports the results from a retrospective study on 26 PMM2-CDG patients. They identify seventeen different pathogenic variants and show that the pathogenic variants affecting the dimerization domain have a significantly higher NPCRS, indicating a more severe clinical outcome. Since PMM2-CDG is a rare disease, a sample size of 26 patients the study can be considered a good starting point. As reported in the discussion,  it will be important to repeat this analysis with more patients.

Author Response

Thank you for your kind evaluation. Indeed as stated in the discussion the analysis needs to be repeated in a larger sample size.

Reviewer 2 Report

In this paper, the authors address the genotype-phenotype correlation in PMM2-CDG by performing a retrospective study involving 26 patients and taking into account both, the symptoms observed and the overall severity of the disease depending on the mutations identified. The latter are categorized in three groups, 1) non-functioning mutations, 2) mutations affecting the stability of the protein and 3) mutations interfering with protein dimerization. The authors show that, while mutations of group 2 correlate with a milder disease course when in combination with a non-functioning allele, mutations of group 3 tend to lead to a more severe evolution.

The study includes only 26 patients in total and of course less in the individual subgroups, which may be seen as a negative point. However, the authors address this point, planning to extend the study when a larger cohort is available. In the meantime, the results obtained still bring interesting information to the reader as to how the localisation of the mutations in the protein correlates with the severity of the disease. This could help predict the course of the disease according to the mutation identified and thus be useful for the follow-up of the patients involved.

Thus, this paper offers novel input on PMM2-CDG. It’s written in a concise and clear way and the discussion emphasises the predictive usefulness of the study but also takes into account its limitations.

Nevertheless, a lot of questions remain open as to how the mutations lead to the phenotypes/severity of the disease. Although it may be too much to include experiments on the molecular aspects of the mutations in this paper – the statistical data indeed seems interesting on its own - some hypothesis/speculation as to what happens at the level of the protein/cell could be added to the discussion.
